# Principal component analysis of alpha-helix deformations in transmembrane proteins

**Alexander Bevacqua[1], Sachit Bakshi[2], Yu Xia[1]***

**1** Department of Bioengineering, McGill University, Montreal, Quebec, Canada, **2** Department of Biomedical Engineering, Boston University, Boston, Massachusetts, United States of America

* brandon.xia@mcgill.ca

**Data Availability Statement:** The data used to pursue our work are held in public repositories: The mpstruc database for Membrane Proteins of Known 3D Structure (https://blanco.biomol.uci. edu/mpstruc/), the Orientations of Proteins in

## Abstract

α-helices are deformable secondary structural components regularly observed in protein folds. The overall flexibility of an α-helix can be resolved into constituent physical deformations such as bending in two orthogonal planes and twisting along the principal axis. We used Principal Component Analysis to identify and quantify the contribution of each of these dominant deformation modes in transmembrane α-helices, extramembrane α-helices, and α-helices in soluble proteins. Using three α-helical samples from Protein Data Bank entries spanning these three cellular contexts, we determined that the relative contributions of these modes towards total deformation are independent of the α-helix's surroundings. This conclusion is supported by the observation that the identities of the top three deformation modes, the scaling behaviours of mode eigenvalues as a function of α-helix length, and the percentage contribution of individual modes on total variance were comparable across all three α-helical samples. These findings highlight that α-helical deformations are independent of cellular location and will prove to be valuable in furthering the development of flexible templates in *de novo* protein design.

## Introduction

### α-helices are deformable bodies

The α-helix is an essential secondary structural component commonly observed in native state protein folds. α-helices are broadly classified as a series of backbone atoms arranged in a right-handed helix with a large dipole moment through backbone carbonyl groups that all point in the same direction. The Ramachandran diagram studies backbone steric clashes and degrees of freedom to conclude on which dihedral angles are most appropriate for the α-helix [1]. The helical geometry is typically specified as having a periodicity of 3.6 residues and a rise of $5.4 \, Å$ per helix turn. Although these parameters are generally used to specify the α-helix, by no means is it an immutable structure. α-helices are flexible bodies, as further evidenced by the variety of helical deformations that are recorded in Protein Data Bank (PDB) submissions [2]. The ability to quantify the deformations of flexible elements in a protein fold is paramount for the development of flexible templates in computational *de novo* protein design.

The earliest computational protein design strategies focused on rigid backbone templates. The atomic coordinates of these templates were fixed to simplify the design process and reduce

Membranes (OPM) Database from the University of Michigan (https://opm.phar.umich.edu/), and the Standard Research Collaboratory for Structural Bioinformatics (RCSB) PDB entries (https://www.rcsb.org/) were used to collect alpha-helical data. Below is a list of all of the accession codes necessary to access our minimal data set: Membrane Proteins (transmembrane alpha-helices, extramembrane alpha-helices): 1a91, 1afo, 1bl8, 1fft, 1h6i, 1j4n, 1jb0, 1jgj, 1kpl, 1kqf, 1l0l, 1l0v, 1l7v, 1ldf, 1lgh, 1m56, 1nek, 1nkz, 1occ, 1okc, 1orq, 1ots, 1p49, 1p7b, 1pv6, 1pw4, 1q16, 1q90, 1rhz, 1rwt, 1su4, 1u19, 1u7g, 1uaz, 1vgo, 1xfh, 1xio, 1yce, 1yew, 1ymg, 1z98, 1zcd, 1zll, 1zoy, 1zrt, 2a65, 2a79, 2ahy, 2b2f, 2b6o, 2bbj, 2bg9, 2bhw, 2bl2, 2bs2, 2d57, 2e74, 2f2b, 2gfp, 2h88, 2h8a, 2hac, 2hi7, 2hyd, 2j58, 2j7a, 2j8c, 2jln, 2jo1, 2jwa, 2k1l, 2k9y, 2kdc, 2kix, 2knc, 2ks1, 2ksd, 2kse, 2ksf, 2ksj, 2l16, 2l35, 2l6x, 2l9u, 2lck, 2lcx, 2lj2, 2ljb, 2lnl, 2lzl, 2lzs, 2m3b, 2m59, 2m6i, 2m6x, 2m8r, 2maw, 2mfr, 2mgy, 2mic, 2mmu, 2mpn, 2n2a, 2n4x, 2nq2, 2nr9, 2nwl, 2o9g, 2oar, 2oau, 2onk, 2q7r, 2qfi, 2qjy, 2qks, 2qts, 2r9r, 2uuh, 2vl0, 2vpz, 2vt4, 2w2e, 2w5j, 2wcd, 2wie, 2wit, 2wjn, 2wlk, 2wsw, 2x2v, 2xtv, 2xut, 2xzb, 2yev, 2yvx, 2z73, 2ziy, 2zjs, 2zt9, 2zw3, 2zxe, 3a7k, 3am6, 3aqp, 3ayf, 3b4r, 3b60, 3b8e, 3b9y, 3beh, 3c02, 3chx, 3d31, 3d9b, 3d9s, 3ddl, 3dh4, 3dhw, 3din, 3dww, 3g5u, 3gd8, 3gia, 3h9v, 3hb3, 3hd6, 3hd7, 3hfx, 3hzq, 3j5p, 3j9p, 3j9t, 3jad, 3jbr, 3jc2, 3jyc, 3k07, 3k3f, 3kcu, 3kg2, 3kly, 3kp9, 3m73, 3mk7, 3mkt, 3mp7, 3ncy, 3nd0, 3o7q, 3org, 3p5n, 3pbl, 3pjz, 3pl9, 3puw, 3q7k, 3qe7, 3qf4, 3qnq, 3rce, 3rfu, 3rhw, 3rko, 3rlb, 3rvy, 3rze, 3s0x, 3s8g, 3syo, 3tds, 3tij, 3tx3, 3ukm, 3um7, 3ux4, 3v3c, 3v5u, 3vou, 3vr8, 3vvn, 3vw7, 3w9i, 3waj, 3wdo, 3wfd, 3wkv, 3wme, 3wo7, 3wu2, 3x29, 3zuy, 4a01, 4a2n, 4aps, 4av3, 4aw6, 4ayt, 4b4a, 4bw5, 4bwz, 4c9g, 4cad, 4cof, 4cz8, 4czb, 4daj, 4djh, 4dji, 4dve, 4dxw, 4ea3, 4ej4, 4ev6, 4ezc, 4f4c, 4f4l, 4f4s, 4g1u, 4g7v, 4gbr, 4gc0, 4gd3, 4grv, 4gx0, 4h33, 4hfi, 4hg6, 4hkr, 4hum, 4huq, 4hyg, 4hyj, 4iar, 4ib4, 4ikv, 4il3, 4iu9, 4j05, 4j72, 4j7c, 4jkv, 4jr9, 4k0e, 4k1c, 4k5y, 4kjs, 4kly, 4kpp, 4l6r, 4lds, 4lz6, 4m48, 4m64, 4mbs, 4mm4, 4mnd, 4mrs, 4ms2, 4mt1, 4myc, 4n7w, 4nef, 4ntj, 4o6m, 4o6y, 4od5, 4or2, 4p6j, 4p6v, 4p79, 4pe5, 4pgr, 4phu, 4pir, 4pl0, 4q2e, 4q65, 4qi1, 4qkc, 4qnc, 4qnd, 4qtn, 4quv, 4r0c, 4rdq, 4rfs, 4ri2, 4rng, 4rp9, 4ry2, 4ryq, 4tll, 4tq4, 4tqu, 4tsy, 4uc1, 4uis, 4us3, 4w6v, 4wd8, 4wgv, 4wis, 4wol, 4x5m, 4xk8, 4xnv, 4xt1, 4xu4, 4xyd, 4y7k, 4yay, 4yb9, 4ybq, 4ymk, 4ymu, 4yzf, 4z34, 4zp0, 4zwj, 4zyo, 5a1s, 5a2n, 5a40, 5aex, 5af1, 5an8, 5ayn, 5azb, 5azd, 5b57, 5c78, 5c8j, 5cfb, 5ctg, 5cxv, 5d0y, 5d3m, 5d91, 5da0, 5dir, 5do7, 5doq, 5dqq, 5dsg, 5dwy, 5e9s, 5egi, 5eh4, 5ek0, 5eke, 5eqg, 5eul, 5ezm, 5fgn, 5fl7,

the combinatorial complexity in searching for an optimal protein fold [3]. Studies done with these fixed templates identified sets of side-chain conformations, known as rotamers, that could build a stable protein core for the *de novo* protein [3]. These protein cores were well-suited for folding by hydrophobic collapse, thereby providing a low-energy structure which could stabilize the surface regions [3]. Although the rigid backbone template is a relatively simple model, it is scrutinized for ignoring backbone flexibility. The superposition of 20 different nuclear magnetic resonance structures of PDB entry 1AEL shows slight positional variations in the backbone atom positions [3]. This implies that rigid templates do not properly balance packing energies and deformation energies [4].

Flexible templates offer more design parameters to refine, which introduces the possibility that these templates can further optimize the free energy of a protein fold, with the drawback of a greater computational complexity. These additional parameters stem from backbone flexibility on the atomic scale and the collective flexible motions of secondary structures. The collective deformations experienced by α-helices can be resolved into individual deformation modes (such as bending and twisting), which from a computational standpoint, represent additional degrees of freedom in the *de novo* protein design process over existing rigid template design studies [5, 6].

## α-helix flexibility is analyzed through constituent deformation modes

α-helix flexibility can be investigated using Principal Component Analysis (PCA) on the atomic coordinates of α-helices collected from the PDB. PCA is a data-driven analysis that can be performed on a sample of static α-helical structures to reveal their principal components. In this context, principal components and deformation modes are interchangeable terms because they both originate from two distinct models (PCA and normal mode analysis) that draw similar conclusions on the flexibility of an α-helix. These modes are each represented by one physical deformation and their individual contribution to the overall deformation of the α-helix is quantified by an eigenvalue ($\lambda$). We illustrate the three dominant principal components exhibited in α-helices in Fig 1.

Previous work identified that the three dominant modes of flexibility from the PCA of α-helices are two bending modes and one twist mode [4]. The two largest eigenvalues capture two nearly degenerate bending modes in two orthogonal planes, which is owed to the approximate cylindrical symmetry of an α-helix [4]. The third largest eigenvalue represents a twisting mode along the principal axis of the α-helix [4]. Each deformation mode has a pair of extreme cases, which are shown individually in each subfigure of Fig 1A–1C, but when these extremes are superimposed, they provide a visual aide on the bounds between which an α-helix may deform (See S1 Fig). The work done by Emberly et al. determined these three dominant deformation modes and studied the scaling behaviour of the eigenvalues as a function of the α-helix length [4]. We aim to expand on that research by elaborating on how the dominant deformation modes and scaling behaviour depend on the location of the α-helix in the cell, namely, whether the protein is surrounded by membrane or aqueous environments.

In the past decades, bioinformaticians struggled with the scarcity of high-resolution structural information of transmembrane proteins [7–9]. The amount of publicly available transmembrane data over time has been tracked by Stephen White and co-workers, where they catalogue high-resolution structures of membrane proteins as part of their mpstruc database [10]. In 2003, at the time of the work completed by Emberly et al. [4], 88 membrane proteins were listed on the mpstruc database [10]. This shortage of data would not have led to a comprehensive and convincing analysis for comparing the deformation modes of α-helices in soluble proteins and membrane proteins. Our work covers three different α-helix types:

5g28, 5gko, 5h1r, 5h35, 5h3o, 5hi9, 5hk1, 5hk7,
5i20, 5i32, 5i6c, 5i6x, 5iji, 5iwk, 5iws, 5j4i, 5jwy,
5jyn, 5k7l, 5kbn, 5khn, 5kte, 5ktf, 5kuf, 5kxi, 5l22,
5lil, 5lki, 5llu, 5lv6, 5lwe, 5lwy, 5lxg, 5m87, 5mkk,
5mpm, 5mrw, 5n6h, 5n6m, 5nik, 5nj3, 5nuo, 5nv9,
5o9h, 5oge, 5oon, 5oqk, 5oqt, 5oyb, 5sv0, 5svj,
5sy1, 5t0o, 5t4d, 5t77, 5tcx, 5tj6, 5tqq, 5tv4, 5twv,
5u1d, 5u6o, 5u73, 5u76, 5uak, 5uj9, 5uld, 5ung,
5uni, 5uz7, 5v4s, 5v6p, 5v7p, 5v8k, 5va1, 5vai,
5vbl, 5vew, 5vkq, 5vkv, 5vms, 5vre, 5vrf, 5w3s,
5w81, 5wiv, 5wj5, 5wpv, 5wqc, 5wua, 5wud, 5wuf,
5x0m, 5x33, 5x5y, 5xam, 5xjj, 5xjy, 5xsy, 5xsz,
5xu1, 5xw6, 5y83, 5ywy, 5z1w, 5z96, 5zbq, 5zih,
5zkp, 5zov, 5zsu, 5zty, 5zug, 5zx5, 6a2j, 6a2w,
6a69, 6a93, 6agf, 6ajf, 6ak3, 6al2, 6aye, 6b3r,
6b5v, 6b85, 6b87, 6bar, 6bbj, 6bcl, 6bd4, 6bml,
6bo8, 6bpq, 6bqr, 6btm, 6by2, 6c08, 6c0v, 6c3o,
6c5w, 6c70, 6c9a, 6caa, 6cc4, 6cfw, 6cjt, 6cm4,
6co7, 6coy, 6cq8, 6cse, 6csm, 6cud, 6d0j, 6d26,
6d3r, 6d79, 6d7w, 6d9z, 6djb, 6dmb, 6dnf, 6drk,
6dt0, 6dvw, 6dw0, 6e0h, 6e10, 6e59, 6ei3, 6eid,
6eu6, 6exs, 6ezn, 6f0k, 6f2g, 6f36, 6ffv, 6fl9, 6fn1,
6fnp, 6g1k, 6g9o, 6gci, 6gct, 6grj, 6gy6, 6h59,
6h7d, 6hjr, 6hzp, 6i6b, 6i9d, 6i9k, 6ibb, 6idp, 6iiu,
6ira, 6irt, 6itc, 6iyx, 6iz4, 6j8e, 6j8g, 6jju, 6jlj, 6jmq,
6jxr, 6k7g, 6kg7, 6kkt, 6kzo, 6m96, 6m97, 6m9t,
6me2, 6me7, 6mgv, 6mho, 6mhq, 6mi7, 6mit,
6mix, 6mjp, 6mqu, 6n3q, 6nbf, 6nf4, 6nf6, 6npl,
6nq2, 6nt3, 6nt5, 6nt6, 6o3c, 6o58, 6o84, 6ob7,
6oce, 6oh3, 6oht, 6oly, 6os9, 6ov2, 6p25, 6pis,
6pw5, 6qim, 6qpc, 6qti, 6qum, 6qzi, 6r3q, 6r4l,
6r7x, 6rko, 6roh, 6rqf, 6rtc, 6rz4, 6rz6, 6s7o, 6s8n,
6sqg, 6tdy, 6u9v, 6uiv, 6ukj, 6uzz, 6v1q, 6v22
Soluble Proteins (alpha-helices in soluble proteins):
1a9r, 1alu, 1ami, 1aro, 1b3r, 1b68, 1bcy, 1cip,
1edu, 1ey3, 1ez3, 1fbm, 1gg2, 1ivj, 1lk2, 1ly1, 1lye,
1mab, 1mdt, 1ob5, 1odc, 1p1x, 1qvn, 1sl2, 1t5e,
1u7r, 1w92, 1wuk, 1xf6, 1xq9, 1yov, 1ysl, 1yte,
1yvh, 1yz6, 1z3z, 1z56, 1z6k, 1zmo, 1zq1, 1zr6,
250l, 2a5v, 2a7x, 2a8y, 2af7, 2al1, 2ald, 2b0j, 2b3y,
2b69, 2clb, 2clm, 2cxi, 2cxn, 2cyp, 2d8d, 2d8e,
2dgd, 2dgm, 2dgn, 2dhq, 2dkd, 2dkj, 2dlc, 2dq3,
2du6, 2e0i, 2e7u, 2e94, 2eey, 2eja, 2ekp, 2ep7,
2eph, 2esf, 2et6, 2ev4, 2ez2, 2f6h, 2f6k, 2f6l, 2f6q,
2fah, 2fba, 2fbw, 2fdw, 2feu, 2fhs, 2fje, 2fl0, 2fq6,
2fym, 2g64, 2g85, 2g9j, 2ggi, 2h31, 2hej, 2hgz,
2hhp, 2hix, 2hmf, 2hy6, 2i14, 2i7x, 2ifc, 2iru, 2isw,
2j4j, 2j5b, 2j9z, 2ja8, 2jd5, 2jgy, 2jib, 2nuw, 2nvl,
2nvq, 2o0b, 2o3k, 2o4j, 2ob2, 2oni, 2p6g, 2pgw,
2pmq, 2q01, 2qfv, 2qmr, 2qna, 2qnv, 2qpp, 2qzo,
2r02, 2r0m, 2r0n, 2r32, 2r8o, 2rah, 2rak, 2rcu,
2rd0, 2rd6, 2rdu, 2rgh, 2rgj, 2rgn, 2rgr, 2rgz, 2rh4,
2rk3, 2rus, 2sbl, 2toh, 2tys, 2uuo, 2uuq, 2ux0,
2uxx, 2uyy, 2uza, 2v0m, 2v0o, 2v0p, 2v0v, 2v1p,
2v1y, 2v29, 2v2e, 2v3w, 2v4m, 2v5j, 2v75, 2v77,
2v7d, 2v8s, 2v9g, 2var, 2vc5, 2vc7, 2vck, 2vd3,
2vdu, 2vig, 2vm6, 2vm8, 2xa7, 2xb6, 2xiq, 2xpx,

transmembrane α-helices, extramembrane α-helices, and α-helices in soluble proteins. We aim to substantiate and validate the conclusions reached by Emberly et al. [4] using a dataset that is over 500% the size of theirs. Furthermore, we expand the study of dominant principal components into several cellular environments to examine how an α-helix's cellular milieu affects the physical deformations it experiences in its native state.

As an α-helix approaches its native state conformation, the total deformation it experiences will be partitioned between bending and twisting. We study this partition using the variance explained by each principal component as a function of the α-helix length across membrane and aqueous environments. If these profiles are similar between cellular environments, then the variance explained by each deformation mode would exclusively rely on α-helix geometry. The variance explained by each principal component as a function of the α-helix length consequently describes an important relationship between the proportion of deformation manifested as bending or twisting, the cellular milieu of the α-helix, and the α-helix length; however, these profiles would not describe differences in α-helical mechanical properties (intensive properties) across cellular milieus. For example, prior work from Bavi et al. used molecular dynamics to estimate the Young's modulus of α-helices from *M. tuberculosis* and *E. coli* homolog mechanosensitive channels [11]. Their work concludes that the Young's modulus from α-helix stretching simulations is higher in a vacuum than it is in water [11], but this result would not describe exactly how variance is partitioned between the constituent modes.

## Transmembrane and soluble proteins have notable similarities and differences

Transmembrane α-helices and α-helices in soluble proteins have different amino acid compositions. The analysis done by Baeza-Delgado et al. on amino acid composition in α-helices revealed that transmembrane α-helices possess glycine and large hydrophobic amino acids such as leucine, valine, isoleucine, and phenylalanine more frequently whereas polar amino acids like glutamate, lysine, asparagine, arginine, and glutamine were less prevalent [8]. Although their study had 792 transmembrane α-helices and 7348 α-helices in soluble proteins compared to our study with 6075 transmembrane α-helices and 6716 α-helices in soluble proteins, our conclusions on the most prevalent amino acid types were the same (S2 Fig).

In a bioinformatic study of the yeast membrane proteome where membrane-embedded transmembrane residues were compared with extramembrane residues, it was concluded that for a fixed degree of residue burial, transmembrane regions evolve 42% more slowly than extramembrane regions using the ratio of the rate of nonsynonymous substitutions to the rate of synonymous substitutions at the DNA level [12]. The transmembrane regions evolve more slowly since the membrane environment imposes greater selective constraint than the aqueous environment surrounding the extramembrane regions [12–14]. Even more, residue evolutionary rate scales in a strong, positive, and linear trend with relative solvent accessibility in both transmembrane and extramembrane regions of membrane proteins [12]. Although extramembrane regions of membrane proteins and soluble proteins have different functional roles, they are both surrounded by an aqueous environment and have similar linear relationships between residue-level evolutionary rate and relative solvent accessibility [12].

Hydrogen bonding is a crucial force in preserving native state transmembrane protein folds. A polar residue in a transmembrane protein is thermodynamically unfavourable unless it is in a hydrogen bonded state as a result of the low dielectric constant of the membrane environment [15]. Transmembrane apolar to polar mutations can lead to non-native hydrogen bonding which can compromise protein function and lead to diseased phenotypes [15]. The glycine-to-arginine mutation alone leads to 4.8% of all transmembrane domain phenotypic

2xr9, 2xrg, 2xv4, 2xvu, 2xy9, 2xzr, 2y0m, 2y7c,
2y8l, 2ypa, 2yq0, 2yxj, 2yxu, 2z6b, 2zas, 2zvu,
2zxm, 2zz9, 3a2h, 3a3k, 3a7d, 3a9z, 3ah8, 3an1,
3azd, 3bf7, 3bxj, 3ch5, 3crk, 3dk8, 3e33, 3e8m,
3epm, 3eqm, 3es7, 3es9, 3etu, 3f2h, 3f3g, 3f46,
3hhp, 3hl8, 3hur, 3i6a, 3i6t, 3k8p, 3kbk, 3kbs,
3kbu, 3kc1, 3kcq, 3kcz, 3kd5, 3kdy, 3ke5, 3kee,
3kfe, 3kfl, 3kgd, 3koz, 3l4k, 3lbo, 3lbs, 3lc6, 3ler,
3let, 3lf9, 3lfv, 3lge, 3lgx, 3lhk, 3lsw, 3lu2, 3lwx,
3lzi, 3m0x, 3m1u, 3m1v, 3m62, 3may, 3mb2,
3mbb, 3mbk, 3mgv, 3mjl, 3mkm, 3moe, 3mrt,
3ms2, 3mtu, 3mve, 3n1z, 3n3b, 3n3m, 3n45,
3n80, 3n94, 3nqj, 3nyd, 3o71, 3odr, 3ovz, 3owa,
3owb, 3owe, 3owf, 3owg, 3owh, 3owi, 3owj, 3oxa,
3oxb, 3oxc, 3oxd, 3oxe, 3oxf, 3oxg, 3oxh, 3oxi,
3oxj, 3oxk, 3oxl, 3oxm, 3oxn, 3oxo, 3oxp, 3oxq,
3oxr, 3oxs, 3oxt, 3oxu, 3oxv, 3oxw, 3oxx, 3oxz,
3oy0, 3oy1, 3oy2, 3oy4, 3oye, 3oym, 3oyn, 3oyo,
3oyp, 3oyq, 3oyr, 3oys, 3oyt, 3oyv, 3oyw, 3oyx,
3oyy, 3oyz, 3oz0, 3oz1, 3oz2, 3oz3, 3oz5, 3oz6,
3oz7, 3ozf, 3ozo, 3ozq, 3ozr, 3ozz, 3p7o, 3p7z,
3p8a, 3p8b, 3p8d, 3p8e, 3p8h, 3p8i, 3p8j, 3p9a,
3p9c, 3p9f, 3p9g, 3p9h, 3p9i, 3p9j, 3p9k, 3p9l,
3p9m, 3p9n, 3p9o, 3p9p, 3p9q, 3p9r, 3p9s, 3p9t,
3p9u, 3p9v, 3p9x, 3p9y, 3p9z, 3pa3, 3pan, 3pao,
3paq, 3pav, 3paw, 3pax, 3pb0, 3pb2, 3pb6, 3pb7,
3pb9, 3pbw, 3pcd, 3pdj, 3pf7, 3pgq, 3pk7, 3plf,
3pm0, 3pny, 3ppi, 3pr2, 3prh, 3psi, 3q0j, 3q24,
3qjx, 3rrf, 3rv6, 3rv7, 3s9v, 3sgw, 3sr7, 3ss7,
3sxp, 3t7v, 3tgm, 3thz, 3tjb, 3tjk, 3tjm, 3tjp, 3tkl,
3tkm, 3tl5, 3tlp, 3tmh, 3tnf, 3tnh, 3tni, 3tnu, 3tq1,
3tso, 3txs, 3u0r, 3u10, 3u1k, 3u2o, 3u2v, 3u3f,
3u5m, 3u5z, 3uas, 3ubr, 3ueh, 3uez, 3uf1, 3ufx,
3ugj, 3uka, 3ul1, 3um8, 3umk, 3up0, 3ur3, 3use,
3ut2, 3ut5, 3uu9, 3uud, 3uul, 3uun, 3uv2, 3uv6,
3uvu, 3uw9, 3uzc, 3uzd, 3v08, 3v0n, 3v2x, 3v3q,
3v4a, 3v4f, 3v4o, 3v4w, 3v5n, 3v5z, 3v6a, 3v72,
3v8b, 3v8d, 3v98, 3v9b, 3v9i, 3vad, 3vbb, 3vd8,
3vec, 3vfc, 3vjs, 3vtk, 3vzd, 3w03, 3w5q, 3w8h,
3wcf, 3wd9, 3wkt, 3woa, 3wod, 3woo, 3wp0,
3wtq, 3zev, 4ari, 4at0, 4at9, 4ayc, 4b82, 4b8l,
4bab, 4bax, 4be8, 4bny, 4boy, 4bql, 4bqq, 4bqr,
4brb, 4bs1, 4buo, 4bwg, 4bwv, 4c08, 4c1u, 4c20,
4c27, 4c2r, 4c2t, 4c31, 4c5c, 4c6b, 4c6o, 4c7x,
4c8b, 4cc5, 4cfg, 4cgz, 4ckl, 4cx3, 4cxx, 4cxy,
4d06, 4d9t, 4dck, 4dcn, 4ddk, 4dfx, 4dnk, 4dzy,
4ehq, 4elj, 4eq5, 4ews, 4f9k, 4fa6, 4fgu, 4fhk, 4fl5,
4flc, 4flh, 4fvm, 4fvx, 4fw9, 4fwj, 4fxf, 4fxo, 4fxs,
4fyt, 4fyx, 4fyz, 4fza, 4fzb, 4fzd, 4fzl, 4fzs, 4fzw,
4g03, 4g0v, 4g1t, 4g27, 4g2d, 4g2k, 4g2v, 4g3m,
4g48, 4g6o, 4g70, 4g75, 4g7l, 4g7o, 4g7r, 4g7t,
4g9q, 4ga8, 4gam, 4gap, 4gbu, 4gfj, 4ghe, 4ghk,
4ghl, 4ghw, 4gi7, 4gic, 4gij, 4gkf, 4gkw, 4gl5, 4glf,
4gmm, 4gnl, 4gnu, 4gnz, 4h2h, 4hb4, 4j3h, 4j5p,
4j8f, 4j9y, 4jgh, 4jp2, 4jsx, 4lv8, 4lv9, 4mhq,
4mmt, 4mmv, 4mo1, 4mx6, 4n9v, 4nkp, 4nsc,
4nyx, 4op1, 4ori, 4p5d, 4p9e, 4pj3, 4pju, 4pk5,

mutations, which is statistically more frequent than its occurrence in soluble proteins [15]. More generally, Partridge et al. determined that residues which participate in hydrogen bonds "are overrepresented as molecular causes of disease when they replace a native [transmembrane domain] residue" [16].

Transmembrane α-helices exhibit structural irregularities more frequently than α-helices in soluble proteins. The standard α-helix is defined in terms of several key metrics including the number of residues per turn (which falls between 3.4 and 4.0) and the rise per residue (between 1.36 Å and 1.76 Å) [17]. α-helix structural irregularities include kinks, the $3_{10}$-helix, and the π-helix [17]. If the local bending angle at a residue within an α-helix is greater than 20°, then the hydrogen bond between residue $i$ and $i$ +4 is broken, and it is consequently called a kinked helix [17]. Hall et al. determined that 44% of transmembrane α-helices had a significant helical kink, with 35% of those kinks caused by proline [18]. The angles of proline-based helical kinks are modulated by proximal serines and threonines [18, 19]. Non-proline kinks were mainly associated with serines and glycines at the center of the kink [7, 18]. In particular, the serine side chain of residue $i$ forms a hydrogen bond with either residue $i$−4 or $i$+4 [7, 18]. The $3_{10}$-helix is a tight-turning and tall α-helix with a periodicity of less than 3.4 residues per helix turn and a rise of greater than 1.76 Å per residue [17]. The π-helix is a wide-turning and short α-helix with a periodicity of greater than 4.0 residues per helix turn and a rise of less than 1.36 Å per residue [17]. Kinks ($K$), kinks associated with tight turns ($K$−$3_{10}$), and kinks associated with wide turns ($K$−$π$) are more frequently observed irregularities in transmembrane α-helices than in α-helices in soluble proteins [17]. More specifically, the ratios ($TM$:$soluble$) are 6:1 for $K$, 9:5 for tight turns, and 11:4 for wide turns [17]. These irregularities are biologically relevant as White et al. show that serine and threonine motifs shape the local structure of transmembrane α-helices through local kinking to improve both solvation and flexibility [20].

In response to the similarities and differences between transmembrane and soluble proteins on a residue-level, we studied the effect of an α-helix's cellular environment on its deformation modes, the scaling behaviour of its eigenvalues, and the contribution of each physical deformation to the overall flexibility of the secondary structure.

## Results and discussion

There are notable comparisons between transmembrane proteins and soluble proteins highlighted by previous research on amino acid propensity, residue-level evolutionary rates, hydrogen bonding, and the frequency of structural irregularities. We investigated the effect of the surrounding environment on the deformation behaviours of transmembrane α-helices, extramembrane α-helices, and α-helices in soluble proteins. As deformable bodies, the flexibility of an α-helix can be quantified through the collective deformations of its residues using Principal Component Analysis (PCA) [4].

### The total deformation of an α-helix can be resolved into deformation modes

$N$ α-helices of a given length ($L$ residues) were collected from PDB entries (See *Methods*). Once the α-helices were structurally aligned, the raw data for PCA comprised of an $N$ by $3L$ matrix of transformed 3D α-carbon atomic coordinates. We decided to use the α-carbon positions instead of all backbone atoms because α-carbon position appropriately captures the geometry of the backbone and to remain consistent with Emberly et al. [4]. Upon performing PCA, the total deformation of the α-helical sample was segmented into constituent modes, with each mode describing a part of the total deformation. The contribution of each mode to the flexibility of an α-helix is quantified with an eigenvalue ($λ$). These eigenvalues measure the

4pk6, 4pm0, 4psl, 4ptf, 4pw8, 4pwl, 4px2, 4pxk, 4pxo, 4pxs, 4py8, 4pyp, 4q6r, 4q7c, 4q9u, 4q9v, 4qet, 4qg2, 4qgk, 4qho, 4qhr, 4qig, 4qij, 4qiw, 4qje, 4qkq, 4qnl, 4qo7, 4qoy, 4qp9, 4qpr, 4qpz, 4qqu, 4qr6, 4qrd, 4qrv, 4qs8, 4qto, 4qu3, 4qu4, 4quw, 4qwt, 4qx6, 4qxm, 4qys, 4qyz, 4r06, 4r0y, 4r0z, 4r1f, 4r1o, 4r1t, 4r28, 4r3n, 4r3u, 4r3z, 4r4m, 4r51, 4r57, 4r5e, 4r5j, 4r5k, 4r6i, 4r6l, 4r6s, 4r75, 4r7j, 4r7y, 4r8a, 4raj, 4rbt, 4rbu, 4rbv, 4rc1, 4rc9, 4rd9, 4req, 4reu, 4rg3, 4rgb, 4rgk, 4rgw, 4rh3, 4rh7, 4rhc, 4rhe, 4rhl, 4rhp, 4rhy, 4ri7, 4rid, 4rit, 4rji, 4rpu, 4rqe, 4rqx, 4rue, 4ruo, 4rxo, 4rxr, 4rxx, 4ry3, 4s1y, 4tk1, 4to6, 4tpp, 4tth, 4tv1, 4tv3, 4twe, 4twv, 4tyh, 4u0s, 4u3z, 4u44, 4u6r, 4u71, 4u72, 4u74, 4uad, 4uda, 4udy, 4uel, 4uf5, 4ufa, 4uhx, 4w2r, 4w4i, 4w4k, 4w4l, 4w51, 4w5k, 4w61, 4wil, 4xks, 4yaa, 4yk9, 4ysw, 4yty, 4z9h, 4za5, 5a0y, 5add, 5akd, 5ake, 5am9, 5ao4, 5az7, 5az8, 5az9, 5bqf, 5bqg, 5bqk, 5bqn, 5bqp, 5bqs, 5br4, 5bxq, 5c2a, 5c2e, 5c2h, 5c32, 5c34, 5cb4, 5ccg, 5cci, 5cmk, 5fh4, 5fnz, 5fo0, 5fo1, 5g04, 5g05, 5g06, 5g09, 5g0w, 5g17, 5hmq, 5hnr, 5hnt, 5hog, 5hpw, 5i08, 5i1u, 5ide, 5ih3, 5ijq, 5iqz, 5jbn, 5jv2, 5kjn, 5kxh, 5ld0, 5lob, 5lun, 5lv4, 5m0s, 5m10, 5ni6, 5nkg, 5r5a, 5ue8, 5urg, 5v2p, 5v2q, 5v5v, 5v97, 5w5d, 5wjb, 5xpp, 5yls, 6a68, 6ale, 6at2, 6cmo, 6di1, 6dlz, 6dmw, 6dyf, 6eni, 6fpo, 6g65, 6gx2, 6jt1, 6jz7, 6jzz, 6k6t, 6k7r, 6k9c, 6k9e, 6ka1, 6kg6, 6krf, 6kri, 6krp, 6ksb, 6kwy, 6kx6, 6l1x, 6l8x, 6lcj, 6ll2, 6lrg, 6m71, 6mur, 6mva, 6mve, 6mxt, 6n4b, 6n92, 6n94, 6n9v, 6nn5, 6nv2, 6o43, 6oru, 6os7, 6oxm, 6oz7, 6p6p, 6pab, 6pam, 6pcz, 6pfz, 6ph1, 6pk2, 6pk5, 6psp, 6pxy, 6py5, 6q4j, 6qhl, 6qjt, 6qu3, 6row, 6rp2, 6s3l, 6s3s, 6skx, 6sxn, 6t3h, 6ta3, 6tj8, 6tqe, 6tt4, 6txw, 6tzj, 6uqw, 6v47, 6vbj, 6vie, 6vjd, 6vmz, 6vnw, 6w1w, 6w2x, 6wok, 6wsa, 6x1q, 6x2m, 6x2x, 6xae, 6xip, 6xzc, 6ycu, 6yry, 6yvt, 6z42, 7odc.

**Funding:** This work was supported by Natural Sciences and Engineering Research Council of Canada grants RGPIN-2019-05952 (YX) and RGPAS-2019-00012 (YX), Canada Foundation for Innovation grants JELF-33732 (YX) and IF-33122 (YX), and Canada Research Chairs program (YX). The funders had no role in study design, data collection and analysis, decision to publish, or preparation of the manuscript.

**Competing interests:** The authors have declared that no competing interests exist.

variance in $Å^2$ captured by an individual deformation mode. The eigenvalues associated with each of the $3L$ principal components were computed for transmembrane α-helices, extramembrane α-helices, and α-helices in soluble proteins in the range $10 \leq L \leq 25$ for a total of 48 sets of eigenvalues.

## The deformation modes have different magnitudes in different cellular milieu

In Fig 2, the ten PCA modes with the largest eigenvalues are presented for α-helices with 18 residues ($L = 18$). Modes #1–3 in Fig 2 represent the three dominant deformation modes that were illustrated in Fig 1: Bend 1, Bend 2, and Twist. The triplet bars for each mode in Fig 2A are included to compare the eigenvalues in the three types of α-helices that we studied. The PCA eigenvalues for $L = 12$, 15, 21, and 24 can be found in S3 Fig. Since α-helices are roughly cylindrical in shape, the two bending modes have similar eigenvalues. This observation is supported by the work done by Emberly et al., in which they also report a nearly degenerate pair of PCA bending modes with nearly identical eigenvalues [4]. Across all three α-helix types in Fig 2A, the twisting mode represented a smaller contribution to the total deformation with the third largest eigenvalue.

The deformation modes that we elucidated from our samples were larger in magnitude (i.e. the eigenvalues were larger) than those published by Emberly et al. [4] in the range of $10 \leq L \leq 25$. This implies that the total variance in each of our α-helical samples were greater than the total variance in their dataset. This is due to the fact that their threshold for accepting potential candidate α-helices (done by selecting unbroken series of residues with dihedral angles $\{\phi, \psi = -50°\pm30°, -50°\pm30°\}$) [4] was more stringent than ours. In other words, their study was more likely than our study to reject α-helices with more extreme deformation types.

On the topic of total variance exhibited by a helical dataset, since there are different physical constraints in the plasma membrane and the cytoplasm due to differences in hydrogen bonding and electrostatic interactions between the two environments, the total variance in helical deformation will be different in each cellular setting. Therefore, for each respective mode in transmembrane α-helices, extramembrane α-helices, and α-helices in soluble proteins, the eigenvalues should not equal one another, and the amplitude of the individual deformation modes cannot be meaningfully compared across different cellular milieus. To address differences in total variance between each dataset, we normalized the eigenvalues by the total variance in their respective datasets as shown in Fig 2B. The resulting percentage of variance explained is a more worthwhile metric to compare since it describes on a percentage basis the way that total deformation is partitioned between constituent modes.

In the range $10 \leq L \leq 25$, focusing on individual deformation modes, we found the eigenvalues between transmembrane α-helices, extramembrane α-helices, and α-helices in soluble proteins were different. This suggests that the eigenvalues of the deformation modes of an α-helix depend on its cellular environment, owing to differences in the physical constraints of these environments. The amplitudes of the α-helical deformation modes rely on the environmental constraints which restrict their deformation. Other metrics such as the helix's scaling behaviour may not necessarily be reliant on these constraints. To investigate this claim further, we studied the scaling behaviour of the three dominant deformation modes.

## α-helices have comparable scaling behaviours, irrespective of cellular environment

The eigenvalues ($\lambda$) of the first three deformation modes were scaled as a function of the α-helix length ($L$) using a power law function ($\lambda \propto L^{\blacksquare}$). The scaling exponents associated with each of the three types of α-helices are summarized in Table 1 (with more details in S1 Table).

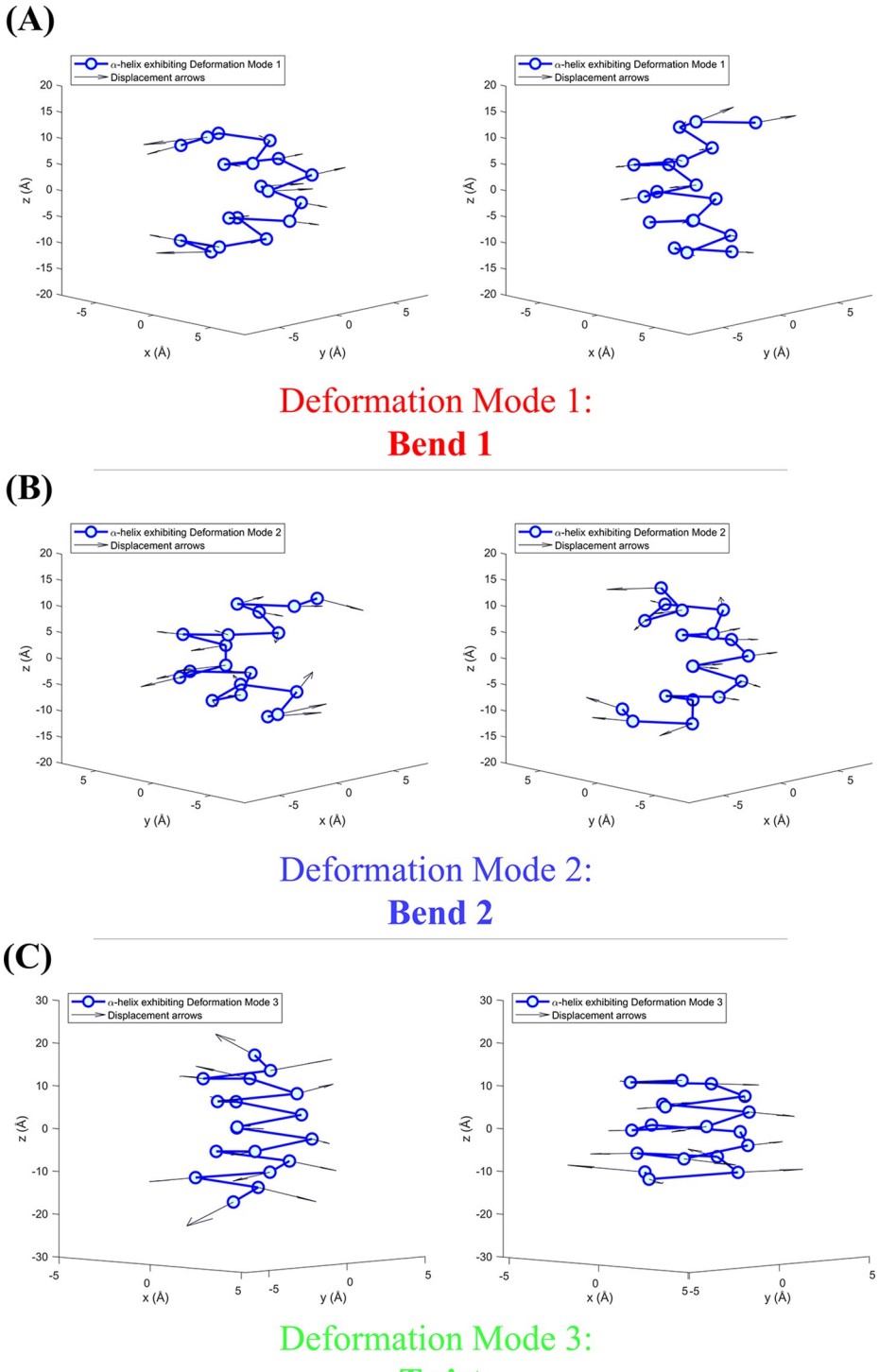

**(A)**

**Deformation Mode 1:**
**Bend 1**

**(B)**

**Deformation Mode 2:**
**Bend 2**

**(C)**

**Deformation Mode 3:**
**Twist**

**Fig 1. The three dominant deformation modes correspond to three physical deformations seen in α-helices with 18 residues ($L$ = 18).** The collections of individual atom displacements on these deformed α-helices lead to individual deformation modes. (A) The first deformation mode, Bend 1, has the largest eigenvalue and it is associated with bending of the α-helix in one plane. (B) The second deformation mode, Bend 2, has the second largest eigenvalue and it is associated with bending of the α-helix in another plane, orthogonal to the first one. (C) The third deformation mode captures the twisting of the α-helix along its principal axis, and it has the third largest eigenvalue. (A)-(C) In each subfigure, the two α-helices are individual helices from the PDB in the transmembrane α-helix dataset that represent the two extreme cases of each deformation mode. The arrows illustrate the displacement vector from each

atom of a standard α-helix (with a periodicity $\Delta\theta$ of 3.6 residues per helix turn, a rise $\Delta z$ of 1.5 Å per residue) to its corresponding atom on the deformed α-helix. The tails of these arrows are all translated to the corresponding atom on the deformed α-helix to more easily illustrate how each atom is pulled under the influence of a particular deformation mode.

The first three columns of entries in Table 1 contain the empirical scaling exponents associated with the eigenvalues of the top three deformation modes in the range $10{\leq}L{\leq}25$. These exponents were calculated by preparing a log-log plot of the α-helix lengths against the PCA mode eigenvalues and identifying the slope of the linear relationship.

For Bend 1 and Bend 2, the scaling exponents were very similar between transmembrane α-helices, extramembrane α-helices, and α-helices in soluble proteins. The scaling exponent of

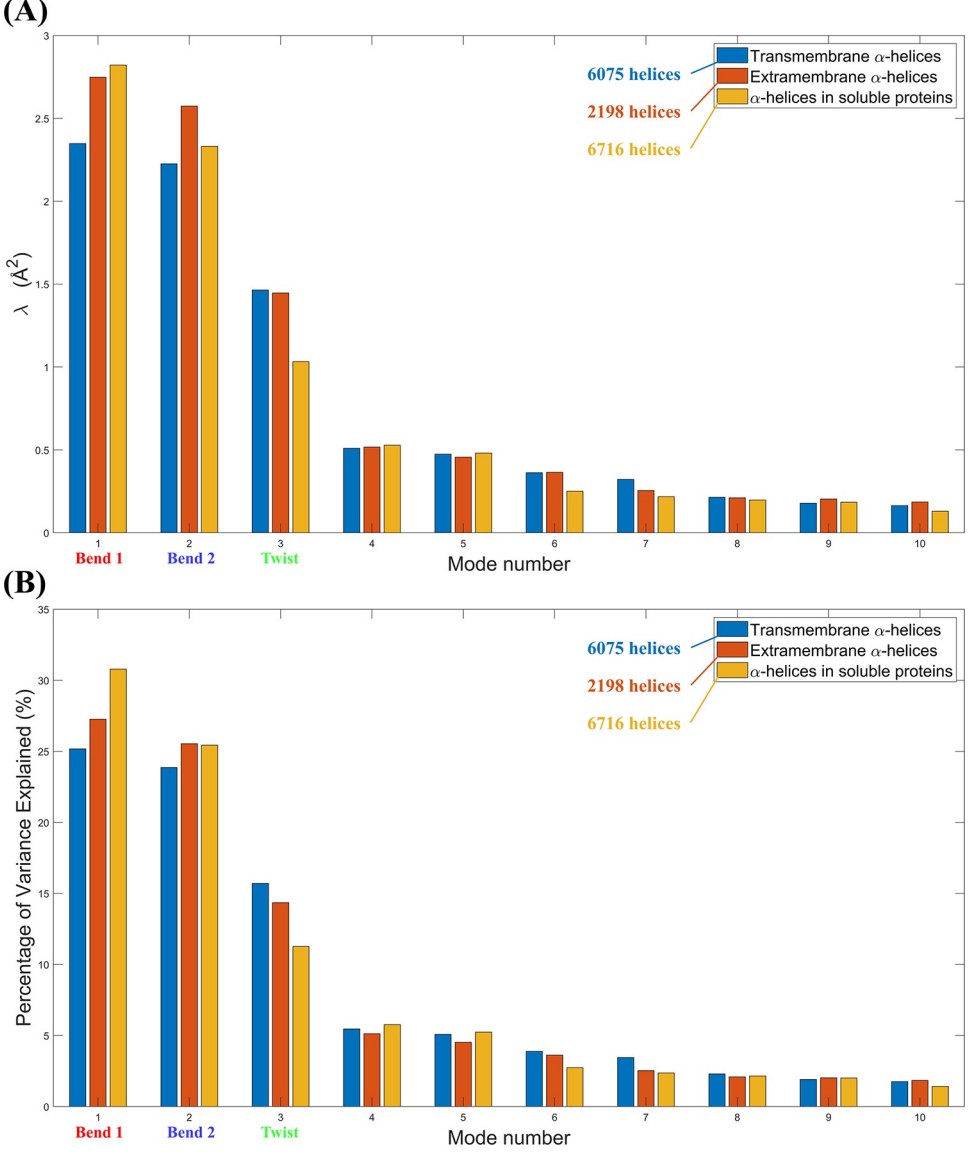

**Fig 2. The ten principal components with the largest eigenvalues ($\lambda$) from 18-residue transmembrane α-helices ($N$ = 6075), extramembrane α-helices ($N$ = 2198), and α-helices in soluble proteins ($N$ = 6716).** (A) The eigenvalues ($\lambda$). (B) The eigenvalues, when normalized by total variance.

**Table 1. The scaling exponents derived from a power law relationship between the eigenvalues ($\lambda$) of the first three deformation modes and the $\alpha$-helix length ($L$).**

| $\lambda \propto L^\blacksquare$ | Transmembrane $\alpha$-helices | Extramembrane $\alpha$-helices | $\alpha$-helices in soluble proteins | $\alpha$-helices in soluble proteins [4] |
|---|---|---|---|---|
| Bend 1 | 3.3 | 3.2 | 3.4 | 4 |
| Bend 2 | 3.6 | 3.5 | 3.6 | 4 |
| Twist | 2.7 | 2.3 | 2.7 | 2 |

Twist is similar between the three different types of helices, especially for the transmembrane $\alpha$-helices and $\alpha$-helices in soluble proteins. Moreover, the scaling exponents of the twisting mode are consistently lower than the scaling exponents of the two bending modes across all three $\alpha$-helix types. The distinction between the bending mode exponents and the twisting mode exponent exists due to the way in which the deformation modes induce displacements away from a mean $\alpha$-helical structure: for bending modes, these displacements increase quadratically with $\alpha$-helix length ($\delta x \approx L^2/R, \lambda_{bend} \propto L^4$) [4]; however, for the twisting mode, these displacements increase linearly with helix length ($\delta x \approx L \delta\theta, \lambda_{twist} \propto L^2$) [4]. In this approach, the scaling of PCA eigenvalues of an $\alpha$-helix was likened to the scaling of a fluctuating elastic rod in thermal equilibrium [4], irrespective of the rod's surrounding environment.

The final column of Table 1 summarizes a key conclusion made by Emberly et al. in their comparisons of the principal components of PCA with the dynamical normal modes of normal mode analysis (NMA) [4]. Unlike PCA, which summarizes a set of related static atomic structures, NMA describes protein dynamics through the collective motions of atoms [21–23]. Emberly et al. used a spring model describing the thermodynamics of a free $\alpha$-helix to determine normal mode eigenvalues representative of the lowest energy deformations and described an inverse relationship between the principal component eigenvalues and the spring constants [4]. In their study, since the top three principal components agreed with the three lowest-energy normal modes, they concluded that the scaling behaviours between PCA modes and normal modes must also match [4]. By approximating an $\alpha$-helix as an elastic rod, they identified that the two bending modes scale with $\lambda_{bend} \propto L^4$ and that the twisting mode scales with $\lambda_{twist} \propto L^2$ [4]. In other words, the data-driven methods of PCA and the fundamental physics arguments of NMA reach the same conclusions on how $\alpha$-helices behave as deformable bodies.

In principle, the results of the NMA should be the same regardless of which environment the elastic rod is located, so the $\alpha$-helical normal modes identified by Emberly et al. are extendable to membrane environments [4]. The results in Table 1 show consistency in PCA scaling behaviour of mode eigenvalues between transmembrane $\alpha$-helices, extramembrane $\alpha$-helices, and $\alpha$-helices in soluble proteins. This is evidence that the way deformation depends on $\alpha$-helix geometry (i.e., scales with helix length) is independent of cellular microenvironment.

## The contribution of each deformation mode as a fraction of total $\alpha$-helix flexibility

Next, we investigated the percentage of contribution made by each deformation mode to the overall flexibility. Since the eigenvalues each measure the variance in $\text{Å}^2$ captured by an individual deformation mode and the total variance was different in each of the three $\alpha$-helical samples that we investigated, it would be worthwhile to normalize the eigenvalues across all three $\alpha$-helical samples as a percentage of their total variance (from all $3L$ deformation modes) for $10 \leq L \leq 25$. Then, eigenvalue trends can be observed independent of the differences in total variance between the three $\alpha$-helix samples.

The eigenvalues of the deformation modes are normalized in Fig 3 to display trends across the principal component number and trends along the $\alpha$-helix length. When comparing

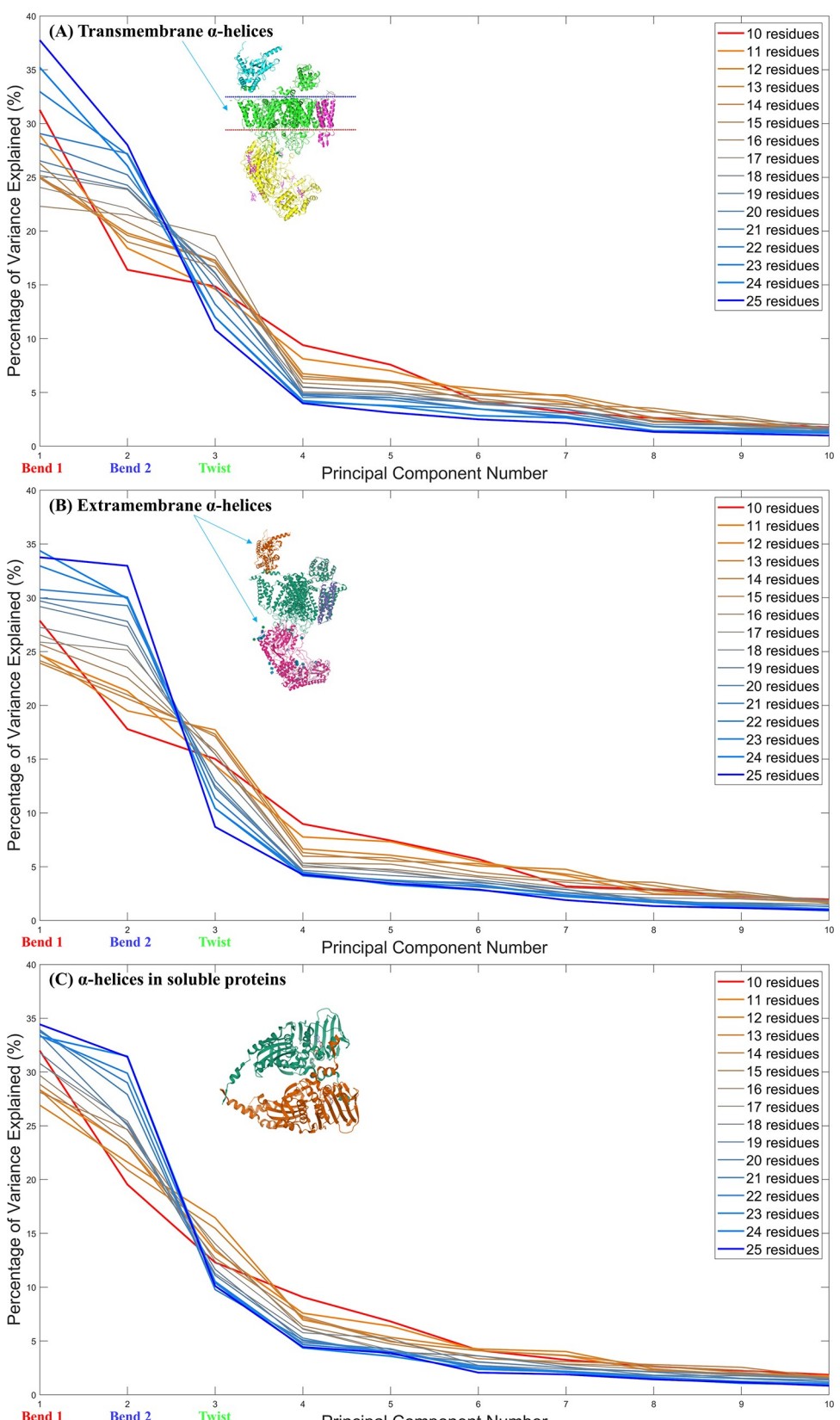

**Fig 3. Each line represents the percentage of total variance explained by the first ten principal components for α-helices of a certain length (L).** Sixteen lines are plotted to illustrate this trend in the range $10 \leq L \leq 25$. The length of the α-helix in question is represented by the colour and thickness of each line. These distributions were plotted for (A) transmembrane α-helices, (B) extramembrane α-helices, and (C) α-helices in soluble proteins. The structures of PDB entries 3JBR [24] and 5AM9 [25] are shown for illustrative purposes.

transmembrane α-helices, extramembrane α-helices, and α-helices in soluble proteins in Fig 3, the collection of sixteen lines in each panel are all generally concave up. By inspection, the blue lines, which describe the relative contribution of each deformation mode for $L = 25$, have a much greater concavity (steeper initial 'slope') than the red lines, which describe the relative contribution of each deformation mode for $L = 10$. This means that the fraction $(\lambda_{Bend1} + \lambda_{Bend2})/\lambda_{Twist}$ is much greater in 25-residue α-helices than in 10-residue α-helices. This follows our intuition well since we expect large, exaggerated bends to hold a greater contribution to the total deformation in the longer α-helices. In fact, the percentage of variance explained by the twisting mode is lower in 25-residue α-helices than in 10-residue α-helices across all three α-helix types shown in Fig 3.

While the fourth and fifth deformation modes are not negligible in magnitude when compared with the three dominant deformation modes, we decided to focus on the first three because they capture the majority of variance explained. This is illustrated more clearly in Fig 4, where we can more closely examine how Bend 1, Bend 2, and Twist–the most prominent physical deformations–contribute the majority of variance explained in each cellular environment.

Following each of the pink lines in Fig 4 from left to right, the summed contributions of the first three principal components describe around 60% of the variance explained for $L = 10$ and the variance explained rises to around 75% as the α-helix length increases to $L = 25$. This observation is invariant to changes in the location of α-helices in the cell. This remarkable similarity between transmembrane α-helices, extramembrane α-helices, and α-helices in soluble proteins is another indication towards α-helix principal components relying primarily, if not solely, on the geometry as opposed to its cellular environment.

The relative importance of the bending modes in explaining the total variance within all three samples increases as the α-helix gets longer as illustrated by the red and blue lines in Fig 4. The relative importance of the twist mode in explaining the total variance within all three samples lowers as the α-helix gets longer as illustrated by the green line in Fig 4. These directional trends match the results of the previous study done by Emberly et al. on 680 α-helices in coiled-coil structures [4]. In these coiled-coiled motifs, once α-helix lengths exceeded 80 residues, higher-order harmonics of the bend mode become lower in energy than the twist mode (i.e. the higher-order harmonics of the bend mode explain a greater percentage of variance than the twist mode) [4]. This means that in α-helices in coiled-coil motifs with lengths greater than 80 residues (and in free α-helices with lengths exceeding 33 residues), the twisting mode will cease to be the third lowest normal mode (and therefore will no longer be the third largest eigenvalue in PCA either as we had represented in Fig 2) [4]. This is consistent with the steady decrease in the percentage of variance explained by the twisting mode in the range $10 \leq L \leq 25$ across all three α-helix types that we observed in Fig 4. The diminishing importance of the twisting mode across all α-helix types as $L$ increases implies that higher-order harmonics of the bending mode will overshadow the twisting mode in longer α-helices regardless of the α-helix's location in the cell. This overshadowing of the twisting mode will rarely be a concern in transmembrane α-helices, and consequently transmembrane protein design since the thickness of the cell membrane imposes a natural constraint on the maximal length of transmembrane α-helices.

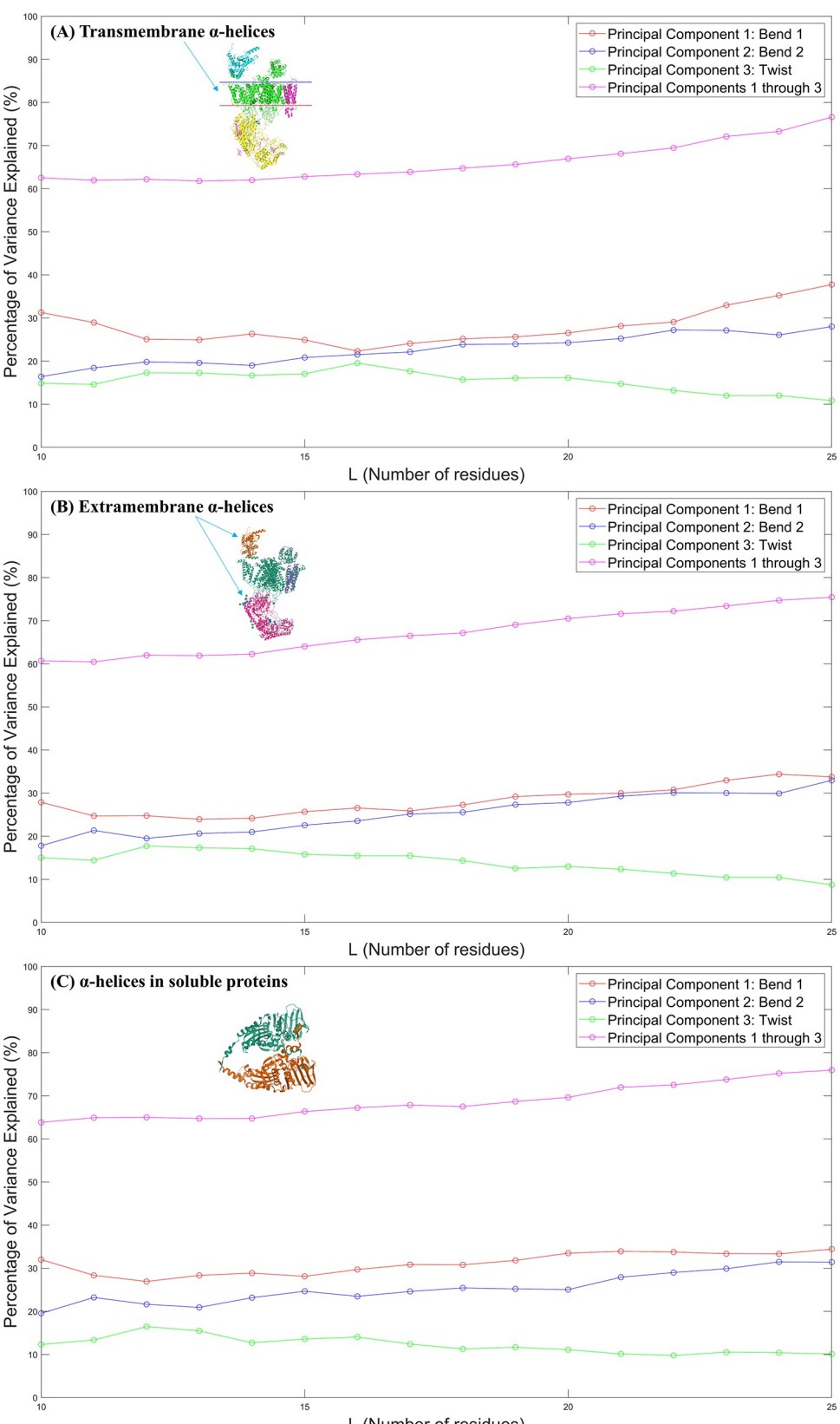

**Fig 4. The percentage of total variance explained by each of the first three principal components individually (red, blue, and green) and combined (pink) for α-helices with helix lengths ($L$) in the range 10≤$L$≤25.** The red, blue, and green lines represent the contributions of Bend 1, Bend 2, and Twist modes respectively towards explaining the total variance. The pink line represents the summed contributions of the first three principal components towards explaining the total variance. These results are plotted for (A) transmembrane α-helices, (B) extramembrane α-helices, and (C) α-helices in soluble proteins. The structures of PDB entries 3JBR [24] and 5AM9 [25] are shown for illustrative purposes.

Returning to the computational work on an α-helix's Young's modulus by Bavi et al., they determined that water acts as a 'lubricant' as the TM1 α-helix in a mechanosensitive channel pore is elongated [11]. At first glance, since the reported Young's modulus of their simulated α-helix is higher in a vacuum than it is in water (i.e., the α-helix is stiffer in a vacuum than in water) [11], it would appear to contradict our conclusion that deformation mode scaling behaviour and percentage of variance explained are independent of cellular surroundings. Deformation modes (including the profiles of variance explained) across cellular milieus cannot be directly compared with an intensive property like Young's modulus. For Bavi et al., the difference in Young's modulus is attributed to changes in the number of hydrogen bonds between the solvent and the helix [11], but for our study, a constant number of native state hydrogen bonds are automatically accounted for in the static PDB structure of each α-helix. Consequently, it is possible to have a lower Young's modulus in an aqueous environment, while also maintaining the same percentage of variance explained profile seen in both a membrane environment and an aqueous environment.

We considered the possibility that the resolution of the protein structures used to pursue our study could affect the deformation mode eigenvalues, scaling behaviour, and percentage of total variance explained that we observe. The average resolution of soluble proteins collected in our study is 2.31 Å and the average resolution of soluble proteins collected in our study is 3.02 Å (see the histograms in S4 Fig). We repeated our analysis on structures within our original three datasets that have a resolution of ≤ 3 Å. The ten largest eigenvalues of 18-residue α-helices across the three datasets in protein structures with a resolution of ≤ 3 Å are presented in S5 Fig. Using these eigenvalues, the scaling exponents (in S2 Table), and the percentage of variance explained by each deformation mode (in S6 and S7 Figs) were calculated. The results of our high-resolution analysis closely match the ones presented in our main study, except for the extramembrane α-helices' scaling behaviour. With a resolution of 3 Å as an upper bound, the extramembrane α-helix dataset shrunk to about 20% of its original size. As presented in S2 Table, this resulted in a Bend 2 scaling exponent of 2.9 (NMA predicts a scaling exponent of 4 for bending modes) and a Twist scaling exponent of 2.1 (NMA predicts a scaling exponent of 2 for the twisting mode).

Future work stemming from our analysis could go in several directions. We decided to use $L$ α-carbons in each α-helix for PCA to remain consistent with Emberly et al. [4] and pursued the assumption that in any one α-helix, if side chain-environment interactions led to some native state structural deformation of the backbone, then it might be manifested in the corresponding α-carbon coordinates that we see in the PDB. It would be worthwhile to include side chain identities in PCA, which would imply that the dataset would need to be segmented by cellular microenvironment, α-helix length, as well as by sequence. This would require a far greater amount of data than is available now. Moreover, in future work, α-helices could be stratified by their degree of solvent exposure, but this would also require more data than is available now, especially for membrane proteins.

In addition to including residue identity and degree of solvent exposure, future analyses could include all α-helix backbone atoms. This would open the possibility of using torsion

angle representations since this approach follows the assumption that bond lengths are invariant. Since the distance between α-carbons is not uniform, this internal representation would not be accurate with the α-carbon dataset we used to pursue this study. Furthermore, an analysis of all α-helix backbone atoms could lead to an improved understanding of how the prevalence of structural irregularities such as kinks between transmembrane α-helices, extramembrane α-helices, and α-helices in soluble proteins depend on α-helix length.

In our analysis, the top three deformation modes are manifested as Bend 1, Bend 2, and Twist specifically because PCA outputs the principal components using an orthogonal basis. We selected PCA as it is considered a data-driven counterpart to NMA [4]. It is possible as future work to analyze the α-helix atomic coordinates using other data-driven approaches such as Independent Component Analysis (ICA), which will not force the components into an orthogonal basis. At the same time, the independent components likely will present the results differently in such a way that they would not be directly comparable to NMA.

## Conclusion

We investigated the relationship between the cellular surroundings of an α-helix and their deformation modes by performing PCA on three α-helical samples representative of three different cellular contexts: transmembrane α-helices, extramembrane α-helices, and α-helices in soluble proteins. Our findings confirmed that for α-helices with lengths in the range of 10–25 residues, the total deformation is described primarily by two nearly degenerate bending modes and a twisting mode. The eigenvalues, which quantify the variance in the sample captured by each individual deformation mode, were calculated across all three cellular milieus and used to study the scaling behaviour of the eigenvalues as a function of the α-helix length using a power law function. The scaling exponents were consistent across the three types of α-helices even though the eigenvalues were not comparable. The independence of deformation mode scaling behaviour on cellular surroundings supports the theory and applicability of normal mode analysis in diverse cellular contexts [4]. The different physical constraints of each cellular environment led to differences in the total variance of each dataset, implying that the amplitudes of individual deformation modes were different across the three different samples. We then studied the contribution of each deformation mode as a fraction of the total deformability in our α-helical samples by plotting mode eigenvalues that were normalized by the total variance of their respective datasets. From these plots, we inferred that the relative contributions of the bending modes and the twisting mode towards the total deformation relied on the length of the α-helix, and not their environment. The similarity between the scaling behaviour and percentage of variance explained profiles of transmembrane α-helices, extramembrane α-helices, and α-helices in soluble proteins can be incorporated in flexible templates in computational protein design to refine the structures of *de novo* transmembrane proteins.

## Methods

667 PDB entries classified as α-helical transmembrane proteins were collected from the mpstruc database for Membrane Proteins of Known 3D Structure [10]. These PDB files have α-helix annotations. Their corresponding entry was collected from the Orientations of Proteins in Membranes (OPM) Database from the University of Michigan [26, 27]. The OPM PDB files modify the Standard Research Collaboratory for Structural Bioinformatics (RCSB) PDB entries by rotating the coordinate system of the 3D atomic coordinates [26, 27]. They set the origin (0,0,0) at the center of the membrane bilayer as illustrated in S8A Fig. The z-axis points to the extracellular space and it is a normal vector with respect to the membrane. The OPM PDB files also include the '½ of bilayer thickness' remark at the top of the file [26, 27].

This reported bilayer thickness was used to determine which α-carbons are located inside the membrane.

RCSB PDB files have α-helix annotation information whereas OPM PDB files have transmembrane region information. When these two pieces of information are brought together, then transmembrane α-helical regions can be properly identified and annotated. Each residue (or more specifically, the α-carbon associated with each residue) of the 667 α-helical transmembrane proteins was annotated as either part of an α-helix, as part of a transmembrane region, as part of a transmembrane α-helix (both), or having no annotation (neither).

Once annotation is complete, the outputted files are then imported into MATLAB for structural alignment. To prepare the input data for PCA, $N$ α-helices of equal amino acid length ($L$) must first be superposed. The goal is to optimally overlay each candidate α-helix (represented as an $L$ by 3 matrix) with the ideal α-helix using only translations and rotations. We parameterized an ideal α-helix with a periodicity $\Delta\theta$ of 3.6 residues per helix turn, a rise $\Delta z$ of 1.5 Å per residue, and a radius of 2.3 Å. Complete details on α-helical superposition are in the *Supporting Information* with accompanying illustrations in S9 Fig.

The PCA function in MATLAB [coeff, score, latent,~,explained,~] = pca(___) was used to identify principal components, calculate their associated eigenvalues, and the percentage of variance explained. This protocol was done sixteen times ($10 \leq L \leq 25$) for transmembrane α-helices to study the scaling relationship of deformation mode eigenvalues as a function of α-helix length. A biplot of orthonormal principal component coefficients (of the $3L$ PCA variables) and principal component scores for each of the $N$ observations were used to identify pairs of extreme observations for each of the first three deformation types: Bend 1, Bend 2, and Twist. These extreme α-helix observations were used for illustrative purposes in Fig 1.

The entire methodology outlined above was repeated for two other types of α-helices: extramembrane α-helices and α-helices in soluble proteins. This was done to verify Emberly et al.'s results [4] on α-helix deformation modes and to highlight any potential differences in α-helix flexibility that would arise from its dependence on the surrounding environment. The 667 PDB entries that were used to collect transmembrane α-helix data were also used to collect extramembrane α-helix data. α-carbon atomic coordinates annotated with 'Alpha Helix' in S8B Fig were used as extramembrane α-helix data for import into MATLAB for superposition as well as for PCA. 959 PDB entries were consulted to acquire the data for α-helices in soluble proteins. Files resembling the one in S8B Fig for soluble proteins were prepared in Python 3, and the data was imported into MATLAB for superposition and PCA as outlined in the above methodology.

Once the main deformation modes of each α-helix type were characterized as shown in Fig 1, the scaling behaviours of each mode for each α-helix type was studied (i.e., the relationships between eigenvalues ($\lambda$) and α-helix length ($L$) were elucidated). The scaling exponents recorded in Table 1 were calculated using a log-log plot of the α-helix lengths ($10 \leq L \leq 25$) against the PCA mode eigenvalues using the Curve Fitting Toolbox in MATLAB. The three dominant deformation modes were inspected individually under a power law function. When the eigenvalue data was fit to the relationship $log(\lambda) = a\ log(L)+b$, the parameter $a$ was the appropriate scaling exponent to fulfill the $\lambda \propto L^{\blacksquare}$ relationship in Table 1.

## Supporting information

**S1 Fig. The three dominant deformation modes seen in α-helices with 18 residues ($L = 18$).** (A)-(C) In each subfigure, α-helix 1 and α-helix 2 are individual helices from the PDB in the transmembrane α-helix dataset. More specifically, they represent the two extreme cases of each deformation mode in the transmembrane α-helix dataset. (TIF)

**S2 Fig. Amino acid distribution representative of 18-residue transmembrane α-helices ($N = 6075$), extramembrane α-helices ($N = 2198$), and α-helices in soluble proteins ($N = 6716$).**
(TIF)

**S3 Fig. The ten principal components with the largest eigenvalues ($\lambda$) from 12-, 15-, 21-, and 24-residue transmembrane α-helices, extramembrane α-helices, and α-helices in soluble proteins.**
(TIF)

**S4 Fig. A pair of normalized histograms presenting the resolution of the structures used to pursue our original analysis.** (A) The membrane protein PDB entries, specifically the transmembrane α-helix and extramembrane α-helix datasets, have an average resolution of 3.02 Å. (B) The soluble protein PDB entries used in this analysis have an average resolution of 2.31 Å.
(TIF)

**S5 Fig. The ten principal components with the largest eigenvalues ($\lambda$) from 18-residue transmembrane α-helices ($N = 2617$), extramembrane α-helices ($N = 428$), and α-helices in soluble proteins ($N = 5360$), all for our analysis of only high-resolution structures ($\leq 3$ Å).** (A) The eigenvalues ($\lambda$). (B) The eigenvalues, when normalized by total variance.
(TIF)

**S6 Fig. Each line represents the percentage of total variance explained by the first ten principal components for α-helices of a certain length ($L$) for our analysis of only high-–resolution structures ($\leq 3$ Å).** Sixteen lines are plotted to illustrate this trend in the range $10 \leq L \leq 25$. The length of the α-helix in question is represented by the colour and thickness of each line. These distributions were plotted for (A) transmembrane α-helices, (B) extramembrane α-helices, and (C) α-helices in soluble proteins.
(TIF)

**S7 Fig. The percentage of total variance explained by each of the first three principal components individually (red, blue, and green) and combined (pink) for α-helices with helix lengths ($L$) in the range $10 \leq L \leq 25$ for our analysis of only high-–resolution structures ($\leq 3$ Å).** The red, blue, and green lines represent the contributions of Bend 1, Bend 2, and Twist modes respectively towards explaining the total variance. The pink line represents the summed contributions of the first three principal components towards explaining the total variance. These results are plotted for (A) transmembrane α-helices, (B) extramembrane α-helices, and (C) α-helices in soluble proteins.
(TIF)

**S8 Fig. An overview of our transmembrane α-helix annotation methods.** (A) A cartoon representation of transformed 3D atomic coordinates in the Orientations of Proteins in Membranes (OPM) Database. When the $|z_{coordinate}| < \frac{1}{2}$ lipid bilayer thickness, the α-carbon is part of a transmembrane region. (B) A piece of an outputted annotation text file: The preprocessed data from the RCSB and OPM PDB files include amino acid identity, residue number, protein subunit, α-carbon coordinates measured in Å, and the appropriate annotations.
(TIF)

**S9 Fig. An overview of our α-helix superposition methods.** (A) The candidate α-helix and the ideal α-helix are not yet optimally superposed. (B) In the first step of superposition, the centroid of the candidate α-helix is translated to the origin. (C) In the second step of superposition, the candidate α-helix is rotated with respect to the ideal α-helix. (D) The displacement

between the z-coordinate of α-carbon 6 in candidate α-helix 3 of the sample and the z-coordinate of α-carbon 6 in the mean α-helix is one of many data points in the raw data for PCA. (E) The raw data for PCA is an $N$ by $3L$ matrix recording the displacements between each atomic coordinate of the transformed candidate α-helix and the corresponding atomic coordinate in the mean α-helix.
(TIF)

**S1 Table. The power law relationship between the eigenvalues ($\lambda$) of the first three deformation modes and the α-helix length ($L$).**
(DOCX)

**S2 Table. The scaling exponents derived from a power law relationship between the eigenvalues ($\lambda$) of the first three deformation modes and the α-helix length ($L$) for our analysis of only high-resolution structures ($\leq$ 3 Å).**
(DOCX)

## Author Contributions

**Conceptualization:** Yu Xia.

**Data curation:** Alexander Bevacqua.

**Formal analysis:** Alexander Bevacqua, Yu Xia.

**Funding acquisition:** Yu Xia.

**Investigation:** Alexander Bevacqua, Sachit Bakshi.

**Methodology:** Alexander Bevacqua.

**Project administration:** Yu Xia.

**Software:** Alexander Bevacqua.

**Supervision:** Yu Xia.

**Validation:** Alexander Bevacqua.

**Visualization:** Alexander Bevacqua.

**Writing – original draft:** Alexander Bevacqua.

**Writing – review & editing:** Alexander Bevacqua, Yu Xia.

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
