## [Decision Letter · Decision Letter 0]

17 Jun 2021

PONE-D-21-09843

Principal component analysis of alpha-helix deformations in transmembrane proteins

PLOS ONE

Dear Dr. Xia,

Thank you for submitting your manuscript to PLOS ONE. After careful consideration, we feel that the manuscript does not fully meet PLOS ONE’s publication criteria as it currently stands. As you will see from the attached review comments, reviewer #1 has pointed out some serious reservations against the publication, and in our opinion, the comments need to be addressed before a decision can be made. Therefore, we invite you to submit a revised version of the manuscript that addresses the points raised during the review process. A further review of the manuscript shall be necessary.

We look forward to receiving your revised manuscript.

Kind regards,

Parag A. Deshpande

Academic Editor

PLOS ONE

Journal Requirements:

Reviewers' comments:

Reviewer's Responses to Questions

**Comments to the Author**

1. Is the manuscript technically sound, and do the data support the conclusions?

Reviewer #1: No

Reviewer #2: Yes

2. Has the statistical analysis been performed appropriately and rigorously? 

Reviewer #1: I Don't Know

Reviewer #2: Yes

3. Have the authors made all data underlying the findings in their manuscript fully available?

Reviewer #1: Yes

Reviewer #2: Yes

4. Is the manuscript presented in an intelligible fashion and written in standard English?

Reviewer #1: Yes

Reviewer #2: Yes

5. Review Comments to the Author

Reviewer #1: The manuscript is technically sound. Principal Component Analysis is a well known mathematic approach for identifying correlation of variables such as collective motions in systems, if the variables are defined as the deviation of catesian coordinates from a reference structure. The reference structure is in this case a 'perfect' a-helix peptide of variable length L. The authors show that the three first principal components correspond to deformations and a twist mode of the helical spine. These results seem to be in good agreement with the lowest energy vibrational modes predicted from NMA (Emberly) .

However, the main conclusion arising from this paper is that the 'flexibility' of alpha-helices , which is described in terms of deformations and twisting modes, does not depend on the environment. That means that an alpha helix behaves similarly in solution and embeded in a membrane. This cannot be the case since the interactions (in particular H-bond, electrostatics, etc) with the environment are different (see discussion on page 7 and on Youngs modulus in page 16). Since the authors only consider the Calpha of the a-helix for their PCA, specific interactions between side chains and environment are not taken into acount. Furthermore, although the amino acid composition of a-helices in different environments is discused in pages 6-8, this property is neglected when using only Calpha for PCA .

Therefore I consider that the data does not support the conclusions regarding the flexibility of the entire alpha helix in different environment.

Furthermore, there are several points that should be further discussed:

1) Fig 2, : Why are only the results for L=18 plotted and futher discussed?, since the eigenvalues computeted for other L values are different (Figure 3). Eigenvalues 4 and 5 are not negligible compared to 3.

2) in the case of soluble proteins , did the authors only select the solvent exposed a-helices , or simply all helices in the PDB were considered for the analysis?

3) A schematic view of Bend 1 and Bend 2 , in the form of structure with arrows describing the displacements would be helpfull. Does the kink of the bending change with the length?

4) What do we learn by scaling the eigenvalues? How are the scaling factors determined?.

5) Caption in Fig 3, should be written in more detail. Protein structures are hardly visible.

6) What are the resolution of the structures used in this study? Are high resolution structures used for the analysis? if not, the fact that the authors do not see any difference in the PC of a-helices in different environments is most likely a result of the inaccuracy of structure determination.

Reviewer #2: The work follows in the footstep on an older study [4] and analyses the eigenvalues of the top 3 values (and others as well) of the PCA of the superimposed helices in different environments.

As a suggestion: it would be interesting to understand if these results are due to the methodology of representing the data: that is running a PCA calculation over helices that are initially superimposed: that the helices are generally placed so that their main axis is aligned implies that the first two eigenvalues will be those perpendicular to this axis (spanning the two dimensions of the plane). If an internal representation was used (e.g., torsion angles) no superposition was needed, and then maybe the main deformation directions were different? Or alternatively, if ICA was used instead of PCA, there wouldn't be the constraint that these are perpendicular to each other.

Minor comments:

(1) It would be helpful if there are more details why the principal components and deformation modes are interchangeable terms (line 66) -- this is described in greater detail in [4], but it would helpful if one is not required to go there.

(2) Does Figure 1 show the average, and the deformed helix (similar to the figure in [4])? in this case, and otherwise, more details are needed on what exactly is shown.

6. PLOS authors have the option to publish the peer review history of their article (what does this mean?). If published, this will include your full peer review and any attached files.

Reviewer #1: No

Reviewer #2: No

---

## [Author Response · Author response to Decision Letter 0]

12 Aug 2021

Please see our Response to Reviewers, where we respond to each point raised in the Decision Letter. Our response is also included below:

RESPONSE TO COMMENTS FROM REVIEWER #1

Reviewer’s Comments: 

The manuscript is technically sound. Principal Component Analysis is a well known mathematic approach for identifying correlation of variables such as collective motions in systems, if the variables are defined as the deviation of catesian coordinates from a reference structure. The reference structure is in this case a 'perfect' a-helix peptide of variable length L. The authors show that the three first principal components correspond to deformations and a twist mode of the helical spine. These results seem to be in good agreement with the lowest energy vibrational modes predicted from NMA (Emberly) .

Author’s Response:

We thank the reviewer for their positive evaluation of our work and for their very insightful comments, which helped us refine our manuscript and the figures which accompany it. We have addressed all the reviewer’s comments as described below.

Reviewer’s Comments: 

However, the main conclusion arising from this paper is that the 'flexibility' of alpha-helices , which is described in terms of deformations and twisting modes, does not depend on the environment. That means that an alpha helix behaves similarly in solution and embeded in a membrane. This cannot be the case since the interactions (in particular H-bond, electrostatics, etc) with the environment are different (see discussion on page 7 and on Youngs modulus in page 16).

Author’s Response:

We thank the reviewer for this very important comment. We completely agree with the reviewer that the flexibility of α-helices depends strongly on the environment, due to large differences in hydrogen bonding, electrostatics, and packing between α-helices in soluble and membrane proteins. Our main conclusion in the paper is that, despite these large differences in the flexibility of α-helices from soluble and membrane proteins, Principal Component Analysis (PCA) reveals that several specific deformation properties of α-helices do not depend on its cellular environment. These include the physical nature of the top three deformation modes (two degenerate bending modes followed by the twisting mode), the percentage of total variance in helical deformation explained by each deformation mode, and the scaling behaviour of the deformation mode over the length of the helix. 

In response to the reviewer’s comment, we have added the following discussions to the manuscript to clarify that there are indeed significant differences in the overall flexibility of α-helices in different environments.

Page 11, Lines 238-248:

“On the topic of total variance exhibited by a helical dataset, since there are different physical constraints in the plasma membrane and the cytoplasm due to differences in hydrogen bonding and electrostatic interactions between the two environments, the total variance in helical deformation will be different in each cellular setting. Therefore, for each respective mode in transmembrane α-helices, extramembrane α-helices, and α-helices in soluble proteins, the eigenvalues should not equal one another, and the amplitude of the individual deformation modes cannot be meaningfully compared across different cellular milieus. To address differences in total variance between each dataset, we normalized the eigenvalues by the total variance in their respective datasets as shown in Fig 2B. The resulting percentage of variance explained is a more worthwhile metric to compare since it describes on a percentage basis the way that total deformation is partitioned between constituent modes.”

Page 12, Lines 250-257:

“In the range 10≤L≤25, focusing on individual deformation modes, we found the eigenvalues between transmembrane α-helices, extramembrane α-helices, and α-helices in soluble proteins were different. This suggests that the eigenvalues of the deformation modes of an α-helix depend on its cellular environment, owing to differences in the physical constraints of these environments. The amplitudes of the α-helical deformation modes rely on the environmental constraints which restrict their deformation. Other metrics such as the helix’s scaling behaviour may not necessarily be reliant on these constraints. To investigate this claim further, we studied the scaling behaviour of the three dominant deformation modes.”

Pages 20-21, Lines 442-454:

“The eigenvalues, which quantify the variance in the sample captured by each individual deformation mode, were calculated across all three cellular milieus and used to study the scaling behaviour of the eigenvalues as a function of the α-helix length using a power law function. The scaling exponents were consistent across the three types of α-helices even though the eigenvalues were not comparable. The independence of deformation mode scaling behaviour on cellular surroundings supports the theory and applicability of normal mode analysis in diverse cellular contexts [4]. The different physical constraints of each cellular environment led to differences in the total variance of each dataset, implying that the amplitudes of individual deformation modes were different across the three different samples. We then studied the contribution of each deformation mode as a fraction of the total deformability in our α-helical samples by plotting mode eigenvalues that were normalized by the total variance of their respective datasets. From these plots, we inferred that the relative contributions of the bending modes and the twisting mode towards the total deformation relied on the length of the α-helix, and not their environment.”

Reviewer’s Comments: 

Since the authors only consider the Calpha of the a-helix for their PCA, specific interactions between side chains and environment are not taken into acount. Furthermore, although the amino acid composition of a-helices in different environments is discused in pages 6-8, this property is neglected when using only Calpha for PCA . Therefore I consider that the data does not support the conclusions regarding the flexibility of the entire alpha helix in different environment.

Author’s Response:

We thank the reviewer for this important comment. We made the decision to only consider the α-carbons of the α-helix for PCA to remain consistent with Emberly et al.’s prior work so that we can compare our results with theirs, and because α-carbon position appropriately captures the geometry of the backbone of the α-helix. Furthermore, we pursued the assumption that if side chain-environment interactions led to some native state structural deformation of the backbone of the α-helix, then it might be manifested in the corresponding α-carbon coordinates. While it is true that presenting the PCA results of only α-carbons lowers the resolution of the analysis when compared with an analysis on all atoms, the approach we took in preparing the data was consistent across all α-helical samples with very different sequences. In response to the reviewer’s comment, we added this as a caveat on 

Page 9, Lines 202-204.

We agree that including side chain atoms’ positions in PCA would lead to a more comprehensive analysis of the interactions between side chains and the environment. However, this is not feasible at this moment due to lack of data. Since PCA requires the same number (and type) of atoms for all structures, the α-helix data will need to be segmented by cellular microenvironment, length, as well as by sequence. This would require a much greater amount of data than is available now.

In response to the reviewer’s comment, we have revised the manuscript to include the following text in the Results and Discussion section (Page 19, Lines 408-415):

“Future work stemming from our analysis could go in several directions. We decided to use L α-carbons in each α-helix for PCA to remain consistent with Emberly et al. [4] and pursued the assumption that in any one α-helix, if side chain-environment interactions led to some native state structural deformation of the backbone, then it might be manifested in the corresponding α-carbon coordinates that we see in the PDB. It would be worthwhile to include side chain identities in PCA, which would imply that the dataset would need to be segmented by cellular microenvironment, α-helix length, as well as by sequence. This would require a far greater amount of data than is available now.”

Reviewer’s Comments: 

Furthermore, there are several points that should be further discussed:

1) Fig 2, : Why are only the results for L=18 plotted and futher discussed?, since the eigenvalues computeted for other L values are different (Figure 3). Eigenvalues 4 and 5 are not negligible compared to 3.

Author’s Response:

We thank the reviewer for these important comments. Fig 2 presents the deformation modes for L = 18 alone and its purpose is to act as an introduction into the topic of PCA. For the sake of brevity, we only included L = 18 in Fig 2 because soon after, we define a metric that we found more important than the deformation mode eigenvalues: the percentage of variance explained by each deformation mode. Fig 3 and Fig 4 present this more important metric in a systematic way, for helices of all lengths (10≤L≤25).

At the same time, we agree with the reviewer that it is important for readers to be able to consult the deformation mode eigenvalue results for other L values. In response to the reviewer’s comments, we have included in the manuscript the deformation mode eigenvalue results for additional values of L (L = 12, 15, 21, and 24) in a new supplementary figure (S3 Fig), which is analogous to Fig 2, and revised the manuscript’s text accordingly (Page 10, Lines 217-218). 

Across all values of L, we decided that the top 3 deformation modes are an appropriate cut-off since they together contribute over 60% to 75% of total variance explained as observed in Fig 4. Furthermore, these modes were studied because they are the same ones studied by Emberly et al. Eigenvalues 4 and 5 could be analyzed in the same ways that were done for eigenvalues 1 through 3 but is beyond the scope of the current study. In response to the reviewer’s comments, we revised the manuscript to include the following text to provide further justification on why we focused on the top three deformation modes even though deformation modes 4 and 5 are not negligible in magnitude compared to the third eigenvalue (Page 16, Lines 336-340): 

“While the fourth and fifth deformation modes are not negligible in magnitude when compared with the three dominant deformation modes, we decided to focus on the first three because they capture the majority of variance explained. This is illustrated more clearly in Fig 4, where we can more closely examine how Bend 1, Bend 2, and Twist – the most prominent physical deformations – contribute the majority of variance explained in each cellular environment.”

Reviewer’s Comments: 

2) in the case of soluble proteins , did the authors only select the solvent exposed a-helices , or simply all helices in the PDB were considered for the analysis?

Author’s Response:

We selected a representative set of α-helices directly from PDB entries of soluble proteins. These α-helices may be either buried or exposed in the soluble protein. Future work may include further stratification of α-helices according to their degree of solvent exposure, but currently there is not enough data for doing so, especially for membrane proteins. In response to the reviewer’s comment, we have revised the manuscript to discuss this direction for future work (Page 19, Lines 415-417).

Reviewer’s Comments: 

3) A schematic view of Bend 1 and Bend 2 , in the form of structure with arrows describing the displacements would be helpfull.

Author’s Response:

We thank the reviewer for this helpful comment. In response to the reviewer’s suggestion, we created a new Fig 1, adding arrows with displacements to each atom. We agree with the reviewer that the new Fig 1 better illustrates to the reader how the collections of individual atom displacements lead to individual deformation modes.

Reviewer’s Comments: 

Does the kink of the bending change with the length?

Author’s Response:

We thank the reviewer for raising this very interesting question. The current study focuses on PCA-based deformation modes (e.g., bending and twisting), rather than on kinks in helices. While a quantitative analysis of kinks is beyond the scope of the current study, it is a worthwhile topic for future work. In response to the reviewer’s comment, we have revised the manuscript to discuss this direction for future work (Page 19, Lines 423-426).

Reviewer’s Comments: 

4) What do we learn by scaling the eigenvalues?

Author’s Response:

Assuming that α-helices are perfectly elastic rods, normal mode analysis (NMA) predicts that the eigenvalue associated with each dynamical normal mode has a characteristic scaling exponent when plotted against helix length. In particular, the eigenvalue associated with the bending mode is predicted to scale quartically with helix length (λ∝L^4), whereas the eigenvalue associated with the twisting mode is predicted to scale quadratically with helix length (λ∝L^2). 

In our current study, we compared the scaling behaviour of the PCA eigenvalues of α-helices in different cellular microenvironments. We find that the scaling exponents of the top three PCA eigenvalues of α-helices are similar across different microenvironments, and they broadly agree with the NMA-based predictions. The similarity between these scaling exponents suggests a homogeny in how the magnitude of each deformation mode scales as a function of α-helix length across all cellular environments. Furthermore, their consistency supports the theory that the principal components exhibited in an α-helix are comparable with NMA-based dynamical normal modes described by Emberly et al..

Reviewer’s Comments: 

How are the scaling factors determined?.

Author’s Response:

The scaling exponents are determined using the Curve Fitting Toolbox in MATLAB as described in Methods (Page 23, Lines 509-514): 

“The scaling exponents recorded in Table 1 were calculated using a log-log plot of the α-helix lengths (10≤L≤25) against the PCA mode eigenvalues using the Curve Fitting Toolbox in MATLAB. The three dominant deformation modes were inspected individually under a power law function. When the eigenvalue data was fit to the relationship log(λ)=a log⁡(L)+b, the parameter a was the appropriate scaling exponent to fulfill the λ∝L^∎ relationship in Table 1.”

Furthermore, additional information is included in Supporting Information. S1 Table and the text beneath that table:

“The power law relationship for transmembrane α-helices, extramembrane α-helices, and α-helices in soluble proteins was determined by establishing the best fit for the parameters a and b in log(λ)=a log⁡(L)+b. The scaling exponent (slope) and intercept are tabulated alongside their 95% confidence intervals in parentheses.”

Reviewer’s Comments: 

5) Caption in Fig 3, should be written in more detail. Protein structures are hardly visible.

Author’s Response:

We appreciate this important feedback and agree with the reviewer’s comments. In response to the reviewer’s comments, we revised the Fig 3 caption to provide more detail on the relationship between the length of the α-helix, and the colour and thickness of each line (Page 15, Lines 329-334; Fig 3):

“Fig 3. Each line represents the percentage of total variance explained by the first ten principal components for α-helices of a certain length (L). Sixteen lines are plotted to illustrate this trend in the range 10≤L≤25. The length of the α-helix in question is represented by the colour and thickness of each line. These distributions were plotted for (A) transmembrane α-helices, (B) extramembrane α-helices, and (C) α-helices in soluble proteins. The structures of PDB entries 3JBR [24] and 5AM9 [25] are shown for illustrative purposes.”

Furthermore, in response to the reviewer’s comments, we increased the size of the images of the protein structures in both Fig 3 and Fig 4.

Reviewer’s Comments: 

6) What are the resolution of the structures used in this study? Are high resolution structures used for the analysis? if not, the fact that the authors do not see any difference in the PC of a-helices in different environments is most likely a result of the inaccuracy of structure determination.

Author’s Response:

We thank the reviewer for this very important comment. In response to the reviewer’s comment, we have revised the manuscript to include a new supplementary figure S4 Fig showing normalized histograms of the resolutions of the structures used in our study for both membrane and soluble proteins. This figure shows that the average resolution of proteins used in this study is 2.31 Å for soluble proteins and 3.02 Å for membrane proteins. 

The reviewer raises an important point that since soluble proteins have, on average, a better resolution than membrane proteins, our results and conclusions could potentially be confounded by the low-resolution structures included in our analysis. To investigate this, we repeated our study using only high-resolution structures for both membrane and soluble proteins with a resolution of ≤ 3 Å. Our results are broadly consistent, and our conclusions remain unchanged. We present these new results using exclusively α-helices from high-resolution structures in the Supporting Information (S5 Fig, S2 Table, S6 Fig, and S7 Fig). 

We added a new part to our discussion (Pages 18-19, Lines 393-406) that describes the high-resolution analysis we pursued: 

“We considered the possibility that the resolution of the protein structures used to pursue our study could affect the deformation mode eigenvalues, scaling behaviour, and percentage of total variance explained that we observe. The average resolution of soluble proteins collected in our study is 2.31 Å and the average resolution of soluble proteins collected in our study is 3.02 Å (see the histograms in S4 Fig). We repeated our analysis on structures within our original three datasets that have a resolution of ≤ 3 Å. The ten largest eigenvalues of 18-residue α-helices across the three datasets in protein structures with a resolution of ≤ 3 Å are presented in S5 Fig. Using these eigenvalues, the scaling exponents (in S2 Table), and the percentage of variance explained by each deformation mode (in S6 Fig and S7 Fig) were calculated. The results of our high-resolution analysis closely match the ones presented in our main study, except for the extramembrane α-helices’ scaling behaviour. With a resolution of 3 Å as an upper bound, the extramembrane α-helix dataset shrunk to about 20% of its original size. As presented in S2 Table, this resulted in a Bend 2 scaling exponent of 2.9 (NMA predicts a scaling exponent of 4 for bending modes) and a Twist scaling exponent of 2.1 (NMA predicts a scaling exponent of 2 for the twisting mode).”

RESPONSE TO COMMENTS FROM REVIEWER #2

Reviewer’s Comments: 

The work follows in the footstep on an older study [4] and analyses the eigenvalues of the top 3 values (and others as well) of the PCA of the superimposed helices in different environments.

Author’s Response:

We thank the reviewer for their positive evaluation of our work and for their helpful suggestions on improving the clarity of our work and on suggesting future directions for our work. We revised our manuscript to address their comments, as described below.

Reviewer’s Comments: 

As a suggestion: it would be interesting to understand if these results are due to the methodology of representing the data: that is running a PCA calculation over helices that are initially superimposed: that the helices are generally placed so that their main axis is aligned implies that the first two eigenvalues will be those perpendicular to this axis (spanning the two dimensions of the plane). If an internal representation was used (e.g., torsion angles) no superposition was needed, and then maybe the main deformation directions were different? Or alternatively, if ICA was used instead of PCA, there wouldn't be the constraint that these are perpendicular to each other.

Author’s Response:

We thank the reviewer for this important suggestion. While the purpose of the current study is to compare the deformation behaviour of α-helices in membrane versus soluble proteins within the analytical framework of Emberly et al., who used Cartesian representation and PCA, we completely agree with the reviewer that torsion angle representation and ICA are important topics for future work. While the current study only uses α-carbons, we believe that the torsion angle representation would be more appropriate for a future analysis done on all backbone atoms since this approach would assume that bond lengths are invariant. In addition, we agree with the reviewer that it would be interesting to apply ICA to our dataset, especially because ICA does not have an orthogonal basis, but at the same time it would be more difficult to directly compare ICA results to normal mode analysis (NMA). In response to the reviewer’s comments, we revised the manuscript to include the following discussion of future work (Page 19, Lines 419-423):

“In addition to including residue identity and degree of solvent exposure, future analyses could include all α-helix backbone atoms. This would open the possibility of using torsion angle representations since this approach follows the assumption that bond lengths are invariant. Since the distance between α-carbons is not uniform, this internal representation would not be accurate with the α-carbon dataset we used to pursue this study.”

Pages 20, Lines 428-434:

“In our analysis, the top three deformation modes are manifested as Bend 1, Bend 2, and Twist specifically because PCA outputs the principal components using an orthogonal basis. We selected PCA as it is considered a data-driven counterpart to NMA [4]. It is possible as future work to analyze the α-helix atomic coordinates using other data-driven approaches such as Independent Component Analysis (ICA), which will not force the components into an orthogonal basis. At the same time, the independent components likely will present the results differently in such a way that they would not be directly comparable to NMA.”

Reviewer’s Comments: 

Minor comments:

(1) It would be helpful if there are more details why the principal components and deformation modes are interchangeable terms (line 66) -- this is described in greater detail in [4], but it would helpful if one is not required to go there.

Author’s Response:

We thank the reviewer for this helpful comment. We had originally written that line to state the interchangeability of the terms ‘principal components’ and ‘deformation modes’ stemming from Emberly et al. [4] so that there would be no doubt in the reader’s mind about this. We agree that we should expand this statement to more clearly justify this. In response to the reviewer’s comment, we have revised the manuscript to provide more detail on the interchangeability between these two terms (Pages 3-4, Lines 65-68): 

“In this context, principal components and deformation modes are interchangeable terms because they both originate from two distinct models (PCA and normal mode analysis) that draw similar conclusions on the flexibility of an α-helix.”

Reviewer’s Comments: 

(2) Does Figure 1 show the average, and the deformed helix (similar to the figure in [4])? in this case, and otherwise, more details are needed on what exactly is shown.

Author’s Response:

This is an excellent point. For the sake of clarity, we should describe in greater detail exactly what α-helix 1 and α-helix 2 mean and where they come from. In each subplot (A)-(C), α-helix 1 and α-helix 2 are individual helices from the PDB in our transmembrane α-helix dataset. They are plotted specifically because they represent the most extreme cases in our dataset of each deformation type, which makes it easier for the reader to see what the principal components should look like. They are here as a visual aide to provide a more intuitive understanding of what the three dominant principal components resolved from an α-helix are. 

Based on Reviewer #1’s suggestion, we decided to create a new Fig 1 and relocate the Fig 1 from the original manuscript to the Supporting Information (S1 Fig). Fig 1, S1 Fig, and their captions reflect the changes described in this response (Page 4, Lines 80-86; Fig 1).

---

## [Decision Letter · Decision Letter 1]

31 Aug 2021

Principal component analysis of alpha-helix deformations in transmembrane proteins

PONE-D-21-09843R1

Dear Dr. Xia,

We’re pleased to inform you that your manuscript has been judged scientifically suitable for publication and will be formally accepted for publication once it meets all outstanding technical requirements.

Kind regards,

Parag A. Deshpande

Academic Editor

PLOS ONE

Additional Editor Comments (optional):

Reviewers' comments:

Reviewer's Responses to Questions

**Comments to the Author**

1. If the authors have adequately addressed your comments raised in a previous round of review and you feel that this manuscript is now acceptable for publication, you may indicate that here to bypass the “Comments to the Author” section, enter your conflict of interest statement in the “Confidential to Editor” section, and submit your "Accept" recommendation.

Reviewer #1: All comments have been addressed

Reviewer #2: All comments have been addressed

2. Is the manuscript technically sound, and do the data support the conclusions?

Reviewer #1: Yes

Reviewer #2: Yes

3. Has the statistical analysis been performed appropriately and rigorously? 

Reviewer #1: Yes

Reviewer #2: N/A

4. Have the authors made all data underlying the findings in their manuscript fully available?

Reviewer #1: No

Reviewer #2: No

5. Is the manuscript presented in an intelligible fashion and written in standard English?

Reviewer #1: Yes

Reviewer #2: Yes

6. Review Comments to the Author

Reviewer #1: As I mentioned before. The authors address all points in a precise manner.

Thus, I recommend the current version of the manuscript for publication

Reviewer #2: (No Response)

7. PLOS authors have the option to publish the peer review history of their article (what does this mean?). If published, this will include your full peer review and any attached files.

Reviewer #1: No

Reviewer #2: No

---

## [Editor Report · Acceptance letter]

6 Sep 2021

PONE-D-21-09843R1 

Principal component analysis of alpha-helix deformations in transmembrane proteins 

Dear Dr. Xia:

I'm pleased to inform you that your manuscript has been deemed suitable for publication in PLOS ONE. Congratulations! Your manuscript is now with our production department. 

Kind regards, 

on behalf of

Dr. Parag A. Deshpande 

Academic Editor

PLOS ONE